# Study of the Mineralogical Composition of an Alumina–Silica Binder System Formed by the Sol–Gel Method

**DOI:** 10.3390/ma16155466

**Published:** 2023-08-04

**Authors:** Lenka Nevřivová, David Zemánek

**Affiliations:** Faculty of Civil Engineering, Brno University of Technology, Veveri 331/95, 602 00 Brno, Czech Republic; nevrivova.l@fce.vutbr.cz

**Keywords:** sol–gel method, refractory, castable, binder, mullite, mineralogical composition

## Abstract

Colloidal bonds are realized by sol–gel technology. The binder system of the refractory castable belongs to the Al_2_O_3_–SiO_2_ binary diagram. Mullite is the most thermally stable mineral in this system. This work was motivated by an attempt to maximize the mullite content in the NCC binder system, because a high content of mullite is a guarantee of the long service life of refractories. Initially, the mineralogical composition of the pure gel was tested after drying and firing at temperatures between 1000 °C and 1600 °C. The behavior of the gel during drying was described. Subsequently, a method of minimizing gel shrinkage during drying was sought. To this aim, fine fillers (microfillers) of alumina and silica were tested. In particular, the reactivity of the microfillers, the ability of the microfillers to react with the sol to form mullite, and the drying shrinkage of the microfiller-doped gel were evaluated. The study showed that the least suitable source of Al_2_O_3_ in terms of its reactivity is tabular corundum, which produces the lowest amount of mullite. The internal structure of the prepared binder system when using different microfillers was described. Based on the results from the second stage of the work, several complete matrixes of the binder system were designed and the degree of their mullitization at different firing temperatures was studied. During this stage, it was shown that the degree of mullitization of the binder system depends mainly on the microsilica content. In the binder system, the maximum mullite content recorded was 76%. The effect of amorphous SiO_2_ on the bulk density and internal structure of the binder system was also described.

## 1. Introduction

The sol–gel bonded no-cement castable (SGBNCC) is a type of refractory castable that was developed to reduce the need for cement in the manufacturing process. The bond obtained from cement and calcium aluminate cement is connected with the development of high strength at temperatures below 1000 °C [1,2]. A general disadvantage of this is that corresponding linings need to be heated up very sensitively, especially during the initial heating. Because the cement contains calcium oxide, there is also a risk of the formation of low melting phases. The presence of calcium oxide in the cement poses a risk of the emergence of low-melting phases [3]. The SGBNCC has many advantages over traditional refractory materials. As such, it has become increasingly popular in the refractory industry and may become a major player in future applications. The use of sol–gel bonding in the preparation of a refractory castable without cement is a relatively new technique and is becoming more and more important in the construction industry due to its advantages over traditional methods.

The sol–gel process is typically carried out by mixing a precursor solution, which contains nanoparticles that make up the solid material and a solvent [4,5]. The sol can then be transformed into gel, either by evaporating the solvent to leave a solid material, or by adding a chemical agent that promotes the crosslinking of the particles to form a three-dimensional network. The resulting solid material, which is made up of a network of interconnected particles, is known as a gel. This gel can be used as a binder in a no-cement refractory castable [6,7,8].

In order to prepare a no-cement castable using the sol–gel method, several important materials are required. The primary inputs are refractory aggregates and a sol–gel binder. The aggregates used can vary depending on the application and the desired properties of the castable, but they are typically made of materials such as alumina, mullite, bauxite, andalusite, corundum, silicon carbide, silicon nitride, or zirconia [9]. The binder is most commonly colloidal silica or alumina; their transformation to gel leads to a “glue effect” where aggregate particles are linked together [10].

The rheological behavior of the material is also important; it should be able to flow easily and fill small voids in the mold without setting too quickly [11]. Studies have found that the addition of surfactants [12,13] or microsilica [14] can improve flowability and adjust setting time.

Drying is an important step in the production of sol–gel bonded castable refractories, as it helps to remove excess water from the mixture, particularly when the gel is forming strong bonds among the particles. The removal of excess water from the mixture is beneficial, particularly when the gel is forming strong bonds among the particles.

The drying process typically begins with a preliminary drying step, in which the castable is left to dry at room temperature for a period of time. The duration of this step depends on the size and shape of the castable, as well as the ambient humidity and temperature [15,16].

After the preliminary drying step, the castable is typically placed in a drying oven, where it is heated at a controlled temperature and humidity. When it is used as shotcrete, the lining temperature increase must be controlled and planned properly. When it is used as a shotcrete, the temperature increase in the lining must be controlled and planned properly. The temperature and humidity used during this step depend on the composition of the castable and the desired properties of the finished product. In general, the temperature of the drying oven is kept low, between 50 and 150 °C, to prevent any chemical reaction or shrinkage in the castable. In general, a longer drying time is needed for larger and thicker pieces of castable than for smaller and thinner pieces of the castable [14,15,16]. The drying of SGBNCC is a crucial step in the production process, and care must be taken to ensure that it is performed correctly and that the finished product has the desired properties.

The durability of SGBNCC is typically better than that of the traditional no-cement castable refractories. This is due to the unique properties of sol–gel binders which provide several advantages over other types of binders.

One of the main advantages of sol–gel binders is that they create very strong bonds among the elements of the refractory aggregate. This improves the mechanical strength of the castable refractory, making it more resistant to abrasion and corrosion. Additionally, because sol–gel binders are made from pure oxides, they are able to withstand high temperatures and thermal shock better than other types of binders [17,18,19,20]. This means that they are less likely to experience cracking or deterioration due to thermal cycling or rapid temperature changes.

The durability of SGBNCC also depends on the composition of the refractory mixture and the operating conditions of the furnace or equipment in which they are used. The quality of the refractory aggregate, its particle size, its shape, and its chemical composition play a crucial role in determining the durability of the castable.

The mineralogical composition of a sol–gel bond can also have a significant influence on its durability [21,22]. The sol–gel bond provides the castable with its strength and durability under high-temperature conditions. The mineralogical composition of the sol–gel bond can affect the properties of the castable in a number of ways, including impacting its thermal expansion coefficient, thermal conductivity, and corrosion resistance [23].

One key factor that can affect the durability of refractory castable is the presence of alumina in the sol–gel bond. Alumina is a highly refractory mineral that is able to withstand very high temperatures, and it is often used in refractory castable to enhance its thermal performance. However, if the concentration of alumina in the sol–gel bond is too high, it can lead to the formation of microcracks in the concrete, which can deteriorate the material properties and reduce its durability over time [24].

Another important mineral that can influence the durability of refractory concrete is silica. Silica, in the form of silica sol, is added to the sol–gel bond. Amorphous nanoparticles can react with alumina particles at elevated temperatures to form mullite, which provides excellent heat properties of the castable [25,26] and it can be obtained by several techniques [27,28,29,30]. Various types of colloidal silica sols, alumina, mullite, spinel, etc. can be used in the preparation of sol–gel bonded refractory castable. Various ions can be used to stabilize these sols [31].

In our research, a silica sol stabilized with Na^+^ cations was selected. Al_2_O_3_ microparticles were added to the mixture as microfillers. Alumina microparticles with different mineralogical and chemical compositions can be used for research. The composition, properties and behavior of the microparticles in the material are primarily influenced by their manufacturing process.

The study deals with a bonding system realized by the sol–gel method applicable in a no-cement castable. The motivation for carrying out this study was the effort to extend the life cycle of a no-cement castable by increasing the mullite content of the binder system and the absence of knowledge of the influence of the fine Al_2_O_3_ particles on the green strengths and especially the influence on the mineralogical composition of the material. A kind of hybrid synthesis of mullite from colloidal SiO_2_ and Al_2_O_3_ microparticles was tested.

The aim of the study was to describe the mineralogical composition of the sol–gel binder system and its behavior during drying and firing at different temperatures. Another objective was to minimize the linear changes in the binder system during drying by using a microfiller and to characterize the effect of the firing temperature and the influence of a microfiller on the binder system, on its internal structure and mineralogical composition in particular.

## 2. Experimental Procedure

### 2.1. Raw Materials and Mixtures, Sample Preparation

Silica sol, an opalescent aqueous dispersion of amorphous silicon dioxide nanoparticles, with designation K1530KD SChem (Ústí nad Labem, Czech Republic) was used in this study for all tested samples (particle concentration—30%, average particle size—15 nm, a negatively charged surface (anionic) stabilized by a low level of alkaline (Na^+^), density—BD = 1.203 g·cm^−3^, pH = 9, viscosity—η = 5 mPas). A 3.5% solution of NH_4_Cl was used as a gelling agent. To achieve matrix composition, which could be used at elevated temperatures, raw materials with high purity were used in this study. Materials were obtained from their producers—tabular alumina (TA) from Almatis (Ludwigshafen, Germany), reactive alumina (RA) from Nabaltec (Schwandorf, Germany), ground alumina (GA) from Nabaltec (Schwandorf, Germany) and silica fume (SF) from RW Silicium (Pocking, Germany).

Tabular alumina (TA) is a completely sintered and thermally stable material. It has a well-developed α-Al_2_O_3_ crystal structure. The content of Al_2_O_3_ is higher than 99%. The powdered alumina oxide from the Bayer process is grinded in a ball mill while the alumina powder balls are made by a disk granulator. The grains are fired in a shaft kiln after drying. The sintering temperature is 1900–1950 °C and fully sintered tabular corundum structure is subsequently obtained [32]. Ground alumina (GA) is finely ground tabular alumina. Reactive alumina (RA) is a fully ground calcined alumina of which a substantial portion (20 to 90%) is made of primary crystals of less than 1 µm. If the boehmite is fired at approximately 1200–1250 °C, the calcined alumina is produced [32,33,34].

In the first stage, the gel was prepared using 500 mL of sol and a gelling agent, 3.5% NH_4_Cl, with a mixing time of 60 s in a polypropylene mold. The transformed gel was subsequently demolded and conditioned at a laboratory temperature of 21 °C. Then, the gel was dried for 24 h at 110 °C in a laboratory drier, and xerogel was obtained. Samples for analysis were prepared by crushing the dry xerogel. Xerogel samples were fired at 600 °C, 800 °C, 1000 °C, 1200 °C, 1400 °C and 1600 °C with a soaking time of 5 h at a maximum temperature and a temperature increase of 4 °C/min.

In the second stage, fine pastes were prepared from the dry raw materials. Chemical composition, mineralogical composition and particle size of fillers are presented in Table 1.

Raw materials were precisely weighed and mixed with silica sol for 15 min with the designed amount of colloidal silica as presented in Table 2 to obtain a fine paste. The consistency of the paste was the same in all cases, which was checked using a Ford cup. Then, the addition of 1.0% of a gelling agent (NH_4_Cl) was carried out with a subsequent mixing for 1 min. The pastes were then casted into silicone molds with dimensions of 10 × 10 × 100 mm, in which they were transformed to gel. Samples were fired at temperatures in a range from 1000 °C to 1600 °C, with research intention to use temperatures realistic in possible applications.

In the third stage, fine pastes containing the designed amount of all fine fillers were prepared. In the following Figure 1, the calculated theoretical Andreasen particle distribution (APD) is shown using coefficient q = 0.23 with a maximal particle size of 1 mm. Six mixtures were prepared with different fused silica content (SF10–SF20) in order to achieve good refractory properties; thus, the maximal limit of SiO_2_ source was limited to 20%. Sample preparation was similar to that of the second stage.

The final mixture compositions are shown in Table 3.

### 2.2. Methods

Chemical composition analysis of the raw materials was performed by wavelength-dispersive X-ray spectroscopy (WDXRF) using the SPECTROSCAN MAKC-GV (Spectron Company, St. Petersburg, Russia) instrument equipped with the QUANTITATIVE ANALYSIS software (version 4.0, Spectron Company, St. Petersburg, Russia). The samples were analyzed in forms of fused beads. Powder X-ray diffraction analysis of the raw materials and engineered aggregates was conducted on a Panalytical Empyrean diffractometer (Panalytical B.V., Almelo, The Netherlands) equipped with a Cu anode, 1-D position-sensitive detector at convention Bragg–Brentano reflection geometry. The setting was the following: step size—0.013° 2θ, time per step—188 s, and angular range 5–80° 2θ. Quantitative phase analysis was performed via the Rietveld method using Panalytical High Score 3 plus software (version 4.8, Panalytical B.V., Almelo, The Netherlands). Quantitative analysis was performed using zincite (ZnO) as an internal standard (10 wt. % per sample).

Scanning electron microscopy with X-ray microanalysis (SEM/EDS) was conducted on gold-coated mechanically broken specimens (for morphological analyses) and on polished carbon-coated thin sections (for chemical microanalyses) using the TESCAN MIRA 3 instrument (Tescan Orsay Holding a.s., Brno, Czech Republic) with the accelerating voltage of 30 kV.

Cold crushing strength (CCS) according to standard EN 993-5:2018 [35] (using machine MEGA 11-600 D-S, Brio Hranice s.r.o., Hranice, Czech Republic) and cold modulus of rupture (CMOR) according to standard EN 993-6:2018 [36] (using machine Testometric M350–20CT, Testometric Co. Ltd., Rochdale, UK) were carried out after drying and after firing. The apparent porosity, water absorption and bulk density were determined by a vacuum water absorption method with subsequent hydrostatic weighing (standard EN 993-1:2018) [37]. Determination of permanent change in dimensions on heating was tested according to standard EN 993-10:2018 [38].

## 3. Results and Discussion

### 3.1. First Stage

The first stage objective of this study was to describe the transformation of gel to xerogel and the mineralogical composition of xerogel after firing. The prepared gel was demolded after 24 h and air-dried in laboratory conditions until the sample weight became steady. Drying of the gel produces xerogel. The initial water content of the material was 179%. During drying, the material behaves very similarly to clay material; Figure 2. As the water content decreases, the material shrinks, which stops at a certain, critical water content, and drying continues without significant shrinkage. The critical water content in our case was 31%. The total linear change in the material during the transformation of the gel into xerogel was −12.8%. The blue curve represents the water drop during the drying process. The black curve describes the linear changes in the material as a result of the reduction in water content. The graph shows that the drying rate (water drop) and the shrinkage rate (length drop) increased up to a water content of 115.6%; see both blue and black curves. From 115.6% to 31.0%, the water drop and length drop were relatively constant, and in the last stage of drying (between 31% and 0% water content) the water drop was detected without significant linear change in the xerogel.

Mineralogical composition of xerogel after drying and firing at selected temperatures is shown in Figure 3. For the quantitative phase determination, zincite (ZnO) was used as the inner standard. The quantitative analysis of the phase composition is presented in Figure 4.

Xerogel mineralogical composition is predominantly composed of a glassy phase, which is signified in XRD diffractograms by a curved background for the 2θ interval 2–30°, as shown in Figure 3—110 °C, 600 °C and 800 °C. Minor crystalline phases of halite (NaCl) and sal ammoniac (NH_4_Cl) were determined for sample S110. Their presence in xerogel treated by the drying process is caused by their crystallization from the solution (halite crystalized due to reaction between free alkalis contained in colloidal silica and free chloride from sal ammoniac, and sal ammoniac recrystallized because it was used as a gelling agent). After firing at 600 °C, these crystallic products are not present in the material due to their decomposition in a range of 300–600 °C; Figure 3.

Further heat treatment leads to the loss of the glassy phase and the crystallization of tridymite and cristobalite. The formation of cristobalite from xerogel was expected at temperatures above 1100 °C as mentioned elsewhere [39]. In our conditions, cristobalite was identified at firing temperatures as low as 1000 °C. As presented in Figure 4, the highest cristobalite content, 84.7%, was determined at 1600 °C, while the tridymite content decreased from its highest content, 32.6%, at 1200 °C to 4.9%. The following Figure 4 shows crystal phase development at the expense of the glass phase.

The samples were examined using SEM, and the results are presented in Figure 5a–c. The dried sample at 110 °C is presented in Figure 5a and shows an amorphous structure with visible cracks caused by the drying process. Figure 5b shows the original structure of the gel, which is still amorphous and without the crystalline phase content. As the temperature increases, tridymite and cristobalite crystallize from that amorphous phase. The higher the temperature, the more cristobalite is formed at the expense of the glass phase and in the temperature above 1200 °C at the expense of tridymite. The apparent porosity of the ceramic body decreases. The effect of sintering on the original gel structure can be seen in Figure 5c.

### 3.2. Second Stage

In the following experiment, six raw materials were selected and combined with colloidal silica to create a fine paste. Figure 6 shows linear changes after firing. Results show that GA, RA1, RA2 samples have linear dependence during the heat treatment with final shrinkage of 5.4–7.2% at 1600 °C on average. The TA sample, due to its raw material fabrication process, performs as expected, and its maximum linear shrinkage peaks at 1600 °C with a value of 2.1%. Test samples containing fumed silica (SF) showed the highest shrinkage. The shrinkage was already 11.8% when fired at 1000 °C. The maximum shrinkage of 22.1% occurred at 1200 °C. As the temperature was increased above 1200 °C, secondary porosity developed, leading to puffing of the ceramic body during firing. After firing at 1600 °C, a shrinkage of 14% was determined (see Figure 6).

Figure 6 demonstrates the evolution of the mineralogical composition of RA1 xerogel after firing at temperatures of up to 1600 °C. At 1000 °C, the material contains only an α corundum and a glassy phase. After firing at 1100 °C, cristobalite appears in the microstructure and crystallizes from the glass phase. When fired above 1400 °C, mullite is formed, which is consistent with [40]. According to Braga, when the precursor has a high degree of homogeneity, the temperature at which mullite formation begins is low. However, when there is heterogeneity, the mullite formation temperature is considerably increased, reaching temperatures above 1400 °C. The SiO_2_ nanoparticles, which originate from the silica sol, react with the Al_2_O_3_ microparticles to form mullite. The amount of mullite formed depends on the SiO_2_ content and on the reactivity of the Al_2_O_3_ particles. The SiO_2_ content was determined by the amount of sol used in the mixture, which was the same for all recipes. Figure 7 shows that the reactivity of the used alumina filler varies. Tabular alumina is the source of the least reactive Al_2_O_3_, while RA1 was found to be the most reactive. The mullite content after firing at 1600 °C for RA1 mixture was 40%, while for TA it was less than 28%.

### 3.3. Third Stage

It was confirmed, in the second stage, that a temperature higher than 1400 °C is required for crystallization of mullite from the melt; Figure 7. For this reason, firing temperatures of 1200 °C, 1400 °C and 1600 °C were used to study the mineralogy in the final part of the study. As the following Figure 8 demonstrates, it was confirmed that mullite is formed by the reaction of SiO_2_ with Al_2_O_3_ only at firing temperatures above 1400 °C. The source of silica is a sol and fused silica. Sources of Al_2_O_3_ are finely ground tabular alumina, reactive alumina and two types of ground alumina.

The last part of the research was focused to the study of the effect of SF on the mulitization of the material.

In the second stage, it was confirmed that TA, as a source of Al_2_O_3_, is the least reactive one of all the raw materials used; see Figure 9a. Therefore, this raw material was progressively replaced by SF. SF was used as the SiO_2_ source. The composition of the mixtures is shown in Table 3. Mullite was proved in the material when fired at 1600 °C, while its presence was excluded when fired at 1400 °C. This corresponds to the results in the second stage.

The presence of amorphous SF in the mixture was confirmed to have a positive effect on mullite formation. The maximum mullite content (78%), maximum mullitization, was achieved at a 20% SF content in the mixture. According to the pattern of dependence of the mullite content on the SF content, Figure 9b, it can be assumed that further increase in the mullite content in the ceramic body cannot be achieved by increasing the dose of SF in the mixture.

A further increase in mullite content in the ceramic body can only be hypothetically achieved by prolonging the isothermal soaking time at 1600 °C or by increasing the firing temperature.

The process of mullitization is related to the bulk density of the material. The higher the mullite content, the lower the bulk density of the material. This dependence is related not only to the specific weight of mullite (3.17 g·cm^−3^) and corundum (4.0 g·cm^−3^), but also to the porosity of the material. At firing temperatures between 1000 and 1400 °C, there is an increase in bulk density, which corresponds to decrease in apparent porosity for all mixtures. At higher firing temperatures, the 10% and 12% SF mixtures follow this trend. The other mixtures, with a higher glass phase (SF) content in the mixture, show a steep decrease in bulk density at firing above 1400 °C, although the apparent porosity still decreases; see Figure 10.

This phenomenon can be explained by the higher sintering rate of the ceramic body; the open pores are closing and the closed porosity is generated, and the ceramic body is puffing. This hypothesis is also confirmed by the evolution of the bulk and apparent density; see Figure 11. As the firing temperature increases, both the bulk and apparent density steadily decrease and the values approach each other. This is due to disappearance of open pores. It can be assumed that at a 25% SF content, the apparent porosity after firing at 1600 °C would be zero.

As shown in Table 4, after firing at 1200 °C, the material contains α-corundum, β-cristobalite and residual, low-temperature quartz. After firing at 1400 °C, the cristobalite content increases slightly at the expense of quartz. The corundum content does not change. After firing at 1600 °C, the material contains only corundum, mullite and a small amount of cristobalite. The amount of formed mullite corresponds to the amount of SiO_2_ available in the melt. The presence of a silica melt is characterized by the development of a microstructure having prismatic mullite crystals. If the resulting product contains a glassy phase, as in our case, then typical long, needle-like, mullite crystals are present [41]. This phenomenon can be seen in the Figure 12c.

The inner structure of the material becomes closed with increasing temperature. At 1000 °C, we observe a puffy structure in which the used silica fume is clearly identifiable (see Figure 12a), the apparent porosity is almost 25%, and the ceramic body contains only corundum, glass phase (SF) and low temperature quartz. After firing at 1200 °C, the glass phase is transformed into cristobalite. The amount of corundum and low-temperature quartz is unchanged. After firing at 1400 °C, the ceramic body shows a melt that closes the open pores; see Figure 12b. The apparent porosity decreases (18%), and the mineralogical composition remains essentially unchanged. After firing at 1600 °C, the mullite crystals of up to 5 µm are visible in the structure; see Figure 12c. The mullite content is 70%, the secondary, closed porosity of the ceramic body is observed; see Figure 9. The apparent porosity decreases to 10%.

The process of mullite crystallization is in accordance with theory. Mullite crystallizes from a melt rich in Al_2_O_3_ and SiO_2_. The viscosity of the melt decreases with increasing temperature, making the melt more liquid and closing smaller pores. As the melt subsequently cools, mullite crystals grow in the melt. Mullite crystallizes from the melt only at temperatures above 1400 °C; see Figure 7 and Figure 9. At lower temperatures, mullite was not detected in the binder system.

## 4. Conclusions

The study investigated the mineralogical composition and the inner structure of the bonding system of the NCC refractory castable realized by the sol–gel bond. The binder system, the bond, consists of a silica sol and of an alumina matrix with particle sizes of up to 10 μm. The aim was to create a binder with the maximum content of mullite, the most thermally stable mineral of the Al_2_O_3_-SiO_2_ binary system.

In the initial part, the length changes in pure SiO_2_ xerogel during drying and its mineralogical composition after firing to different temperatures were described. The formation of cristobalite and tridymite was confirmed already at a firing temperature of 1000 °C.

In the second part of the study, the reactivity of the alumina matrix was tested with SiO_2_ nanoparticles to form mullite. Tabular corundum was the least reactive source of Al_2_O_3_. For all matrixes, the mullite was shown to crystallize at temperatures above 1400 °C. The maximum mullite content was 40%.

In conclusion, a fine-grained matrix with an optimal granulometric curve was designed, also incorporating SF, which is another source of SiO_2_ required for the formation of mullite. The mullite content is affected by a slight increase in the amorphous SiO_2_ content, but the bulk density of the material is significantly reduced.

## Figures and Tables

**Figure 1 materials-16-05466-f001:**
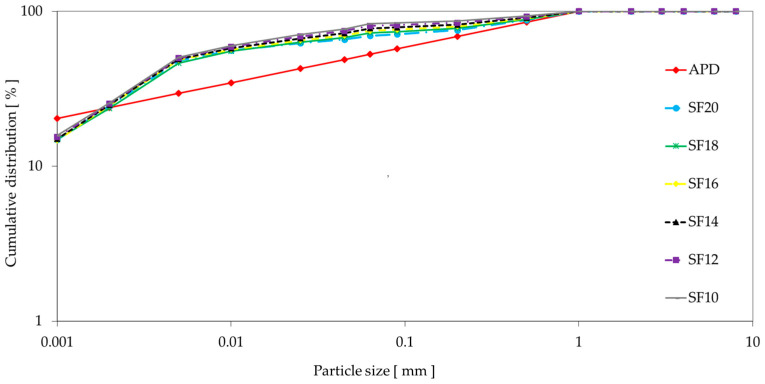
Particle size distribution of prepared mixtures.

**Figure 2 materials-16-05466-f002:**
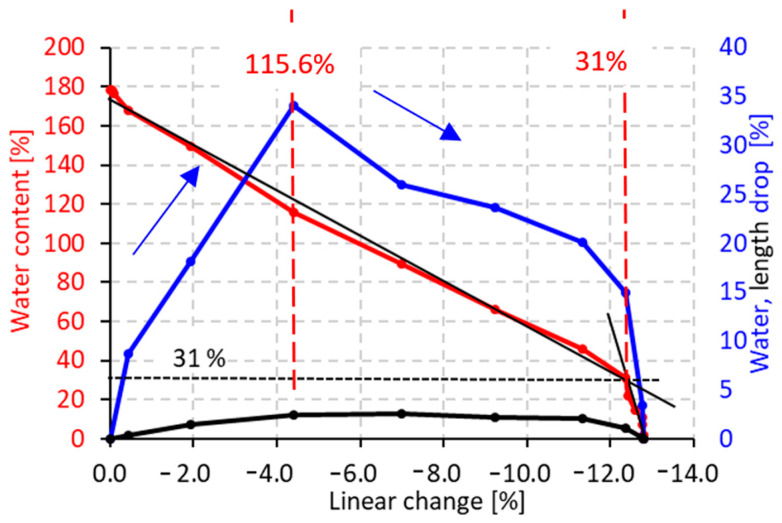
Gel to xerogel transformation process—linear change during the drying process, effect of water content on linear change—red curve, moisture drop as a function of linear change—blue curve, length drop as a function of linear change—black curve.

**Figure 3 materials-16-05466-f003:**
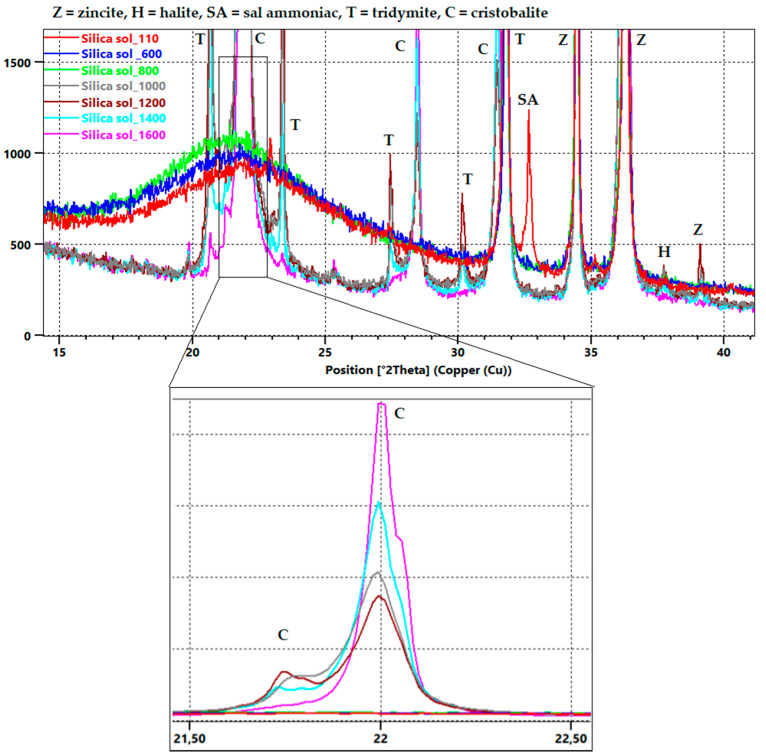
XRD diffractograms with focus on 15–40° 2θ section of xerogel after drying and firing at different temperatures with zoomed area 21.5–22.5° 2θ.

**Figure 4 materials-16-05466-f004:**
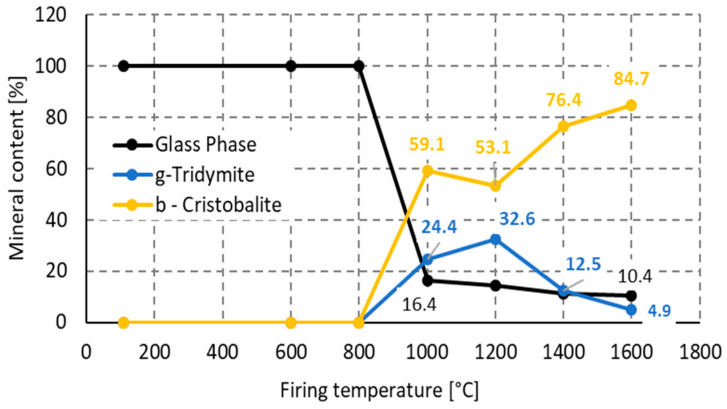
Firing temperature influence to tridymite, cristobalite and glass phase content in ceramic body.

**Figure 5 materials-16-05466-f005:**
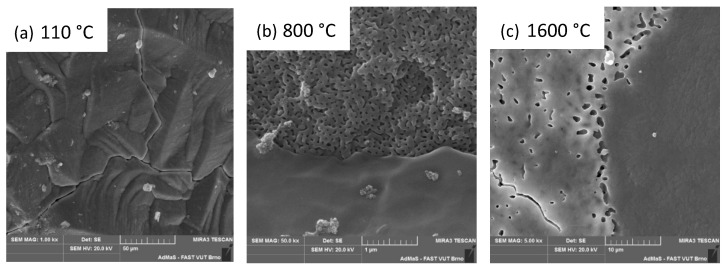
Firing temperature influence to inner structure of the xerogel.

**Figure 6 materials-16-05466-f006:**
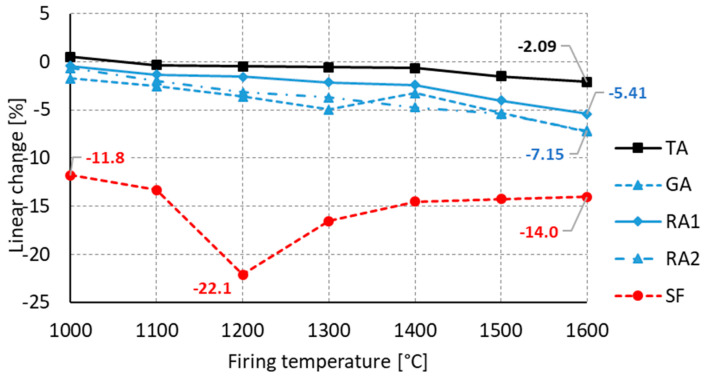
Firing temperature influence to linear change in ceramic body.

**Figure 7 materials-16-05466-f007:**
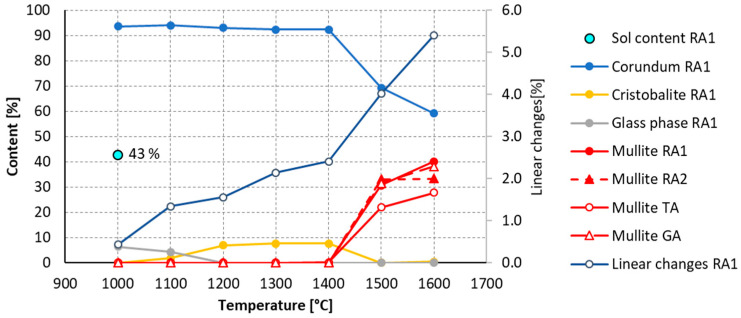
Evolution of mineralogical composition in RA1 material and the effect of filler on the mullite content of ceramic body after firing.

**Figure 8 materials-16-05466-f008:**
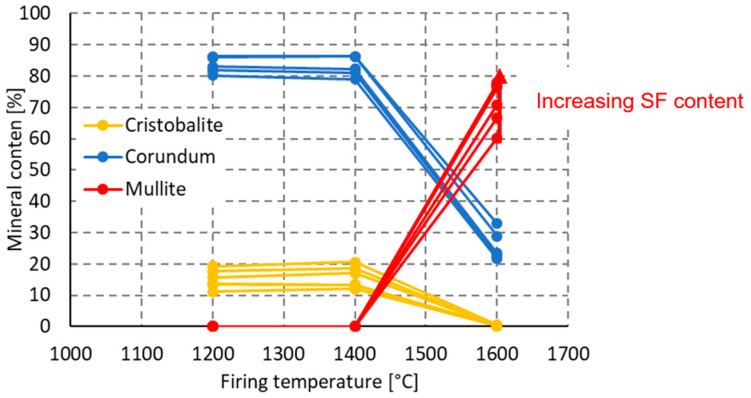
The effect of firing temperature on the mullite, corundum and cristobalite content of ceramic body.

**Figure 9 materials-16-05466-f009:**
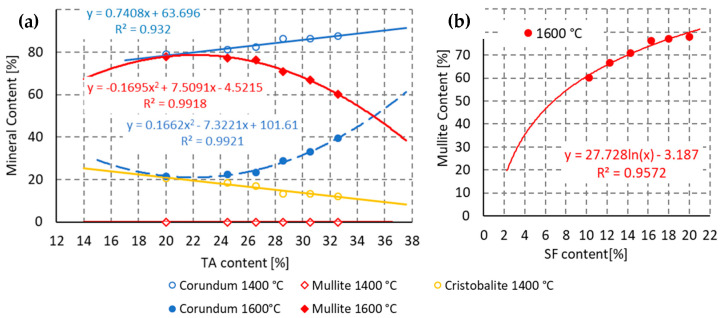
Evolution of mineralogical composition in RA1 material and the effect of matrix on the mullite content of ceramic body after firing. (**a**) Tabular alumina, (**b**) Silica fume.

**Figure 10 materials-16-05466-f010:**
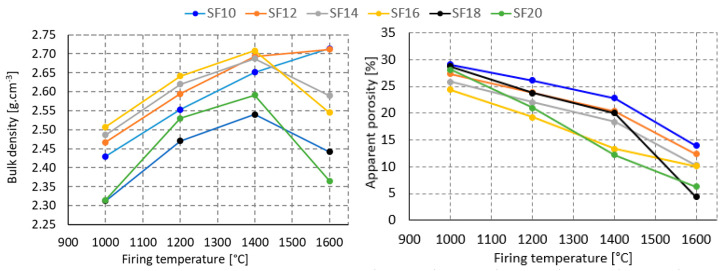
Firing temperature effect on bulk density and apparent porosity of material.

**Figure 11 materials-16-05466-f011:**
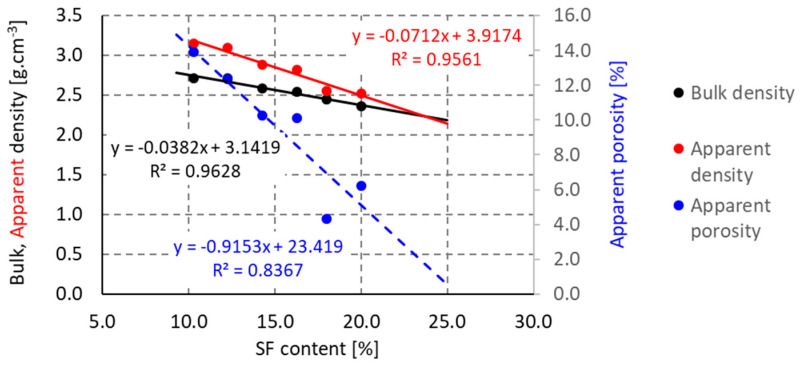
Effect of silica fume content on apparent porosity at firing temperature of 1600 °C.

**Figure 12 materials-16-05466-f012:**
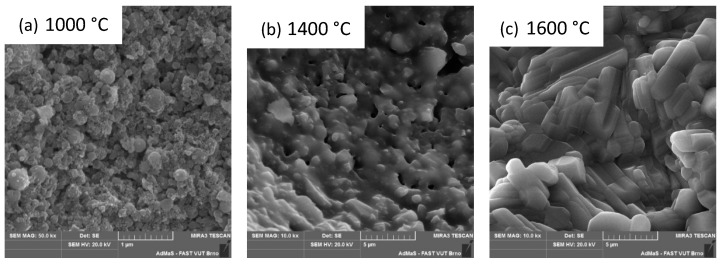
Fire temperature effect on the binder system inner structure, SF content 14%.

**Table 1 materials-16-05466-t001:** Raw material properties.

Chemical Composition	TA	GA	RA1	RA2	SF
SiO_2_	0.09	0.80	0.16	0.12	98.46
Al_2_O_3_	99.55	98.35	99.45	99.53	0.04
TiO_2_	–	0.03	–	–	0.01
Fe_2_O_3_	0.01	0.09	–	–	0.07
CaO	–	0.22	0.01	0.01	0.62
MgO	–	0.23	0.14	0.17	0.04
K_2_O	–	0.03	0.04	0.08	0.55
Na_2_O	0.45	0.25	0.20	0.09	0.01
Phase composition					
Quartz	–	–	–	–	0.26
Corundum	94.98	98.90	100	100	–
Andalusite	–	–	–	–	–
Diaoyudaoite	5.02	1.10	–	–	–
Amorphous phase	–	–	–	–	99.74
Particle size (μm)	0.5–10	1.0–10	0.5–10	1.0–10	0.15–0.2

**Table 2 materials-16-05466-t002:** Sol content of fine pastes in wt. %.

Raw Material	Sol (wt. %)	Gelling Time (s)
TA	38.7	240
GA	42.2	230
RA1	41.7	180
RA2	39.0	180
SF	120.0	120

**Table 3 materials-16-05466-t003:** Fine matrix composition, compounds in wt. %.

Component (wt. %)	SF10	SF12	SF14	SF16	SF18	SF20
TA	32.6	30.6	28.6	26.6	24.5	20.0
GA	21.1	19.9	17.9	15.9	26.0	20.0
RA1	18.6	17.9	17.9	17.9	16.0	20.0
RA2	17.4	19.4	21.4	23.4	15.5	20.0
SF	10.3	12.3	14.3	16.3	18.0	20.0

**Table 4 materials-16-05466-t004:** Mineralogical composition of xerogels after firing at different temperature, compounds in wt. %.

Mineral Content (%)										
	1200 °C			1400 °C			1600 °C			
Fine Matrix	Q	C	Cn	Q	C	Cn	Q	C	Cn	M
SF10	0.5	11.1	88.4	0.3	12	87.7	0	0.3	39.4	60.2
SF12	0.3	13.5	86.2	0.3	13.4	86.3	0	0.3	32.9	66.7
SF14	0.4	13.7	86	0.3	13.4	86.3	0	0.2	28.9	70.9
SF16	1.2	15.8	83	0.6	17	82.3	0	0.2	23.4	76.4
SF18	0.5	17.6	81.9	0.4	18.5	81.1	0	0.4	22.5	77.2
SF20	0.6	19.3	80.2	0.4	20.6	79.1	0	0.5	21.6	77.9

Legend: Q—low temperature Quartz; C—Cristobalite; Cn—Corundum, M—Mullite.

## Data Availability

The data presented in this paper are available upon request from the corresponding author.

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
