# Peer review of "Study of the Mineralogical Composition of an Alumina–Silica Binder System Formed by the Sol–Gel Method"

_materials, 2023, doi:10.3390/ma16155466_

Round 1

Reviewer 1 Report

see the attachment

minor editing

Author Response

Dear reviewer,

Thank you very much for your time and fruitful comments. We accepted some of our suggestions and recommendations and changed in our manuscript. Here are our answers and explanations.

On behalf of both authors

David Zemánek

(response is marked R)

  1. Usually, the abstract is aimed at telling the readers what this research have done, but not introduce the basic material. Therefore, I think the author should improve it by condensing the current abstract down to one paragraph, in which the new abstract should include the information of what the author have done, what result the author have obtained.

R: Has been improved.

  1. I think the introduction should be improved by adding other researchers’ results, including the material's properties and performance, but not just what they have done.

R: Has been improved.

  1. All chemical formulas in the whole manuscript should be corrected for their number subscripts, such as NH4Cl in the abstract.

R: Has been improved.

  1. In Figure 11, the author should mark out a~c as for what.

R: Has been improved.

  1. In Figure 11, it’s clear that the below SEM is the local enlarge view of upper SEM images. The author should mark out which zone have been enlarged in the upper SEM image.

R: We don´t find it so important, so we kept it without change.

  1. It’s obvious that the author gives some explanations of the SEM images in Figure 11 below it. But the author does not explain why this phenomenon.

R: Has been improved.

Reviewer 2 Report

In this paper, the mineralogical composition of an alumina-silica binder system formed by the sol-gel method was investigated. Some comments need to be addressed before the manuscript is considered for publication.

1) In introduction, the motivation and important (difference) of this work should be further presented.

2) It’s better to put the section 2.1 (Methods) after the section 2.2 (Raw Materials and Mixtures, Sample preparation).

3) Why did authors select silica sand as raw materials?

4) What does “Grain size” in Table 1 refer to?

5) Please provide the detailed particle grading of components in Table 3.

6) What does “length loss” mean in Fig. 1?

7) It’s better to add the number of PDF files for each phase in Fig. 2.

8) How to quantitative analysis of the glass phase?

9) It’s better to change the unit of density as g/cm3.

10) Please provide the functions of fitting data in Figs. 8 and 10. Try to analyze the theoretical significance.

Author Response

Dear reviewer,

Thank you very much for your time and fruitful comments. We accepted some of our suggestions and recommendations and changed in our manuscript. Here are our answers and explanations.

On behalf of both authors

David Zemánek

(response is marked R)

  • In introduction, the motivation and important (difference) of this work should be further presented.

R: Has been improved.

  • It’s better to put the section 2.1 (Methods) after the section 2.2 (Raw Materials and Mixtures, Sample preparation).

R: Accepted, improved.

  • Why did authors select silica sand as raw materials?

R: We used silica sand as source of SiO2, but this raw material was deleted from manuscript as other reviewer suggested.

  • What does “Grain size” in Table 1 refer to?

R: Probably you meant in Table 2, and it was further moved from Table 2 to Table 1. 

  • Please provide the detailed particle grading of components in Table 3.

R: Particle grading was measured.

  • What does “length loss” mean in Fig. 1?

R: „Length loss” means negative linear change after firing, it was changed in our article to length drop.

  • It’s better to add the number of PDF files for each phase in Fig. 2.

R: Sorry, we don´t understand your suggestion, but Figure 2 was improved.

  • How to quantitative analysis of the glass phase?

R: Analysis was using Rietveld quantitative analysis using zincite (ZnO) as an internal standard (10 wt.% per sample). Added to the Section 2.1.

  • It’s better to change the unit of density as g/cm3.

R: Accepted, improved.

  • Please provide the functions of fitting data in Figs. 8 and 10. Try to analyze the theoretical significance.

R: Accepted, improved.

Reviewer 3 Report

Abstract should be written in more general way, without detailed experimental data. Please. rewrite abstract.

Introduction

Replace sol-gel bonded no-cement castable (SGBNC) with (SGBNCC).

Sentence The sol-gel process is a method for producing solid materials from small molecules [4, 5]. is too general, please give another description corresponding to the level of the readers of the Journal. This description is as in Wikipedia!

Readers are familiar with sol gel method, rewrite this part,  or dellite section

The sol-gel process is a method for producing solid materials from small molecules [4, 5]. The sol-gel process is a "wet" chemical technique used for the synthesis of a variety of materials, such as ceramics, glasses, and composites.

Add at the end of Introduction part aim of the paper. use 2-3 sentences.

Experimental is well written.

3. Results and Discussion: Fig 1. Please, add legend as it is given at other figures, describing red, blue and black curves.

Figure 4a. explanation is missing, please, add text.

Figure 10.  If density is decreasing, it is expected that porosity is increasing, but it is not the case for your study. Please, explain.

Literature, Introduction: please add references from 2023, 2022.

Author Response

Dear reviewer,

Thank you very much for your time and fruitful comments. We accepted some of our suggestions and recommendations and changed in our manuscript. Here are our answers and explanations.

On behalf of both authors

David Zemánek

(response is marked R)

Abstract should be written in more general way, without detailed experimental data. Please. rewrite abstract.

R: Accepted, improved.

Introduction

Replace sol-gel bonded no-cement castable (SGBNC) with (SGBNCC

R: Accepted, replaced.

Sentence The sol-gel process is a method for producing solid materials from small molecules [4, 5]. is too general, please give another description corresponding to the level of the readers of the Journal. This description is as in Wikipedia!

R: Accepted, deleted.

Readers are familiar with sol gel method, rewrite this part,  or dellite section

The sol-gel process is a method for producing solid materials from small molecules [4, 5]. The sol-gel process is a "wet" chemical technique used for the synthesis of a variety of materials, such as ceramics, glasses, and composites.

R: Accepted, replaced.

Add at the end of Introduction part aim of the paper. use 2-3 sentences.

R: Accepted, improved.

Experimental is well written.

  1. Results and Discussion: Fig 1. Please, add legend as it is given at other figures, describing red, blue and black curves.

R: Accepted, improved.

Figure 4a. explanation is missing, please, add text.

R: Accepted, improved.

Figure 10.  If density is decreasing, it is expected that porosity is increasing, but it is not the case for your study. Please, explain.

R: Accepted, improved, explaned in the text. Read paragrafs above Figure 9 and Figure 10.

Literature, Introduction: please add references from 2023, 2022.

R: Accepted, suitable 2022 reference added [16].

Reviewer 4 Report

The authors begin the article with a rather lengthy introduction, where they give perhaps redundant data regarding the description of tabular alumina, brown fused alumina, etc. However, in the Introduction, the authors do not state the purpose of their study. At the same time, the Conclusion begins with a goal statement, followed by a listing of what has been done. In general, the introduction looks very lengthy and broad.

The numbering of references in the introduction is broken: 19 is followed by 35.

Figure 1 looks overburdened and difficult for the reader to analyze. It is recommended to divide it into several figures or provide a more thorough analysis, particularly the difference between loss length and length drop.

Figure 3 provides the same data as Table 3. Why is there a repetition of information?

When discussing Figure 4, the authors start right away with 4b, ignoring the description of 4a. In the same paragraph, speaking of apparent porosity, the authors do not provide any specific data.

In section 3.2 the authors say "..., which is associated with a volume increase, see Figure 5.". However, Figure 5 represents Llinear change, not Volume change. The authors could analyze the change in density, or show the kinetics of the crystal structure and lattice parameters, especially since they already have the necessary XRD patterns. This would make the conclusions presented more substantiated.

In section 3.3 the authors say "Mixtures were designed so that the granulometry of the matrix corresponded to the optimum granulometric curve according to Fuller" - what is this based on? How is this confirmed?

In all figures, there is no indication of the standard deviation.

In general, the paper creates the impression of a report on the work done, but not a scientific article. The Results and Discussion section contains only one (!) literary reference, which makes all the conclusions of the authors unfounded and unconfirmed. I believe that this work cannot be published in its current form and requires serious revision.

Author Response

Dear reviewer,

Thank you very much for your time and fruitful comments. We accepted some of our suggestions and recommendations and changed in our manuscript. Here are our answers and explanations.

On behalf of both authors

David Zemánek

(response is marked R)

The authors begin the article with a rather lengthy introduction, where they give perhaps redundant data regarding the description of tabular alumina, brown fused alumina, etc

R: Was partly deleteted and partly replaced to 2.2.

However, in the Introduction, the authors do not state the purpose of their study.

R: Added in introduction - last two paragrafs.

At the same time, the Conclusion begins with a goal statement, followed by a listing of what has been done. In general, the introduction looks very lengthy and broad.

R: We agree with your comment, but we wanted to summarize this topic broadly.

The numbering of references in the introduction is broken: 19 is followed by 35.

R: Accepted, improved.

Figure 1 looks overburdened and difficult for the reader to analyze. It is recommended to divide it into several figures or provide a more thorough analysis, particularly the difference between loss length and length drop.

R: Accepted, improved.

Figure 3 provides the same data as Table 3. Why is there a repetition of information?

R: Accepted, deleted.

When discussing Figure 4, the authors start right away with 4b, ignoring the description of 4a. In the same paragraph, speaking of apparent porosity, the authors do not provide any specific data.

R: Accepted, improved.

In section 3.2 the authors say "..., which is associated with a volume increase, see Figure 5.". However, Figure 5 represents Llinear change, not Volume change. The authors could analyze the change in density, or show the kinetics of the crystal structure and lattice parameters, especially since they already have the necessary XRD patterns. This would make the conclusions presented more substantiated.

R: This part was deleted as other Reviewer suggested to remove silica sand from the raw materials, so not relevant at the moment.

In section 3.3 the authors say "Mixtures were designed so that the granulometry of the matrix corresponded to the optimum granulometric curve according to Fuller" - what is this based on? How is this confirmed?

R: We are truly sorry for our mistake. Not Fuller, but Andreasen particle packing model was used.

In all figures, there is no indication of the standard deviation. 

R: We find it not relevant for Figures, where mineralogical composition is presented. This may be considered for Figure 5 and 9.

In general, the paper creates the impression of a report on the work done, but not a scientific article. The Results and Discussion section contains only one (!) literary reference which makes all the conclusions of the authors unfounded and unconfirmed. I believe that this work cannot be published in its current form and requires serious revision. 

R: Accepted, improved.

Round 2

Reviewer 4 Report

The authors have made corrections according to the comments.